# Insights into the Influence of Signal Peptide on the Enzymatic Properties of Alginate Lyase AlyI1 with Removal Effect on *Pseudomonas aeruginosa* Biofilm

**DOI:** 10.3390/md20120753

**Published:** 2022-11-29

**Authors:** Ming-Jing Zhang, Shuai-Ting Yun, Xiao-Chen Wang, Li-Yang Peng, Chuan Dou, Yan-Xia Zhou

**Affiliations:** 1Marine College, Shandong University, Weihai 264209, China; 2Shangdong Kelun Pharmaceutical Co., Ltd., Bingzhou 256600, China

**Keywords:** alginate lyase, *Pseudomonas aeruginosa* biofilm, signal peptide, salt activated

## Abstract

Most reports on signal peptides focus on their ability to affect the normal folding of proteins, thereby affecting their secreted expression, while few studies on its effects on enzymatic properties were published. Therefore, biochemical characterization and comparison of alginate lyase rALYI1/rALYI1-1 (rALYI1: without signal peptides; rALYI1-1:with signal peptides) were conducted in our study, and the results showed that the signal peptide affected the biochemical properties, especially in temperature and pH. rALYI1 (32.15 kDa) belonging to polysaccharide lyase family 7 was cloned from sea-cucumber-gut bacterium *Tamlana* sp. I1. The optimum temperature of both rALYI1 and rALYI1-1 was 40 °C, but the former had a wider optimum temperature range and better thermal stability. The optimum pH of rALYI1 and rALYI1-1 were 7.6 and 8.6, respectively. The former was more stable and acid resistant. Noticeably, rALYI1 was a salt-activated enzyme and displayed remarkable salt tolerance. Alginate, an essential polysaccharide in algae and *Pseudomonas aeruginosa* biofilms, is composed of α-L-guluronate and β-D-mannuronate. It is also found in our study that rALYI1 is also effective in removing mature biofilms compared with controls. In conclusion, the signal peptide affects several biochemical properties of the enzyme, and alginate lyase rALYI1 may be an effective method for inhibiting biofilm formation of *Pseudomonas aeruginosa.*

## 1. Introduction

Signal peptides (SP) are first hypothesized and proposed by Gunter Blobel and David Sabatini in 1971. In 1991, the Nobel Prize in Physiology or Medicine is awarded to Blobel for his discovery of SP [1]. The vast majority of secretory proteins in all domains of life carry a short peptide at their N-terminal end, called SP [2], which is generally composed of 16–30 amino acid residues.

Prokaryotic hosts can be used for the expression of recombinant proteins but problems arise in the large-scale production of recombinant proteins, such as inclusion body formation, and there is also the potential for endogenous proteins to interfere with protein folding. These factors may lead to a loss of protein amount [3]. More than 90% of the secreted proteins in *Escherichia coli* are SP dependent [4]. In addition, a large number of human diseases are associated with SP, and several diseases were associated with SP mutations and damage to human proteins. SP can also be used as a biomarker for the diagnosis of cardiovascular diseases [5]. Altering the SP can improve the secretion efficiency of recombinant lysosomal enzymes, allowing better clinical application of lysosomal enzymes [6]. In addition to the above roles, SP has other applications, including determining protein-folding status, influencing downstream transmembrane behavior and N-terminal glycosylation, and has a potential vaccine candidate application. SP has also been shown to influence heterologous expression in *E. coli*, where it can trap large amounts of proteins in the cell and influence enzyme expression and purification [7].

Alginate is composed of β-D-mannuronate (M) and α-L-guluronate (G) polymerized through 1,4-glycosidic bonds, which forms three modes: poly-G blocks, poly-M blocks, and poly-MG blocks [8]. Alginate lyases can be depolymerized by alginate lyases from various sources including marine algae, fungi and viruses, marine and terrestrial bacteria, marine mollusks, and echinoderms [9,10,11,12,13]. Among these, alginate lyases isolated from marine bacteria are the more extensively investigated. According to the carbohydrate-active enzyme (CAZy) database, alginate lyases belong to 14 different polysaccharide lyases families and the largest alginate lyase family is the PL7 family. Alginate lyases are classified as exo- or endo-alginate enzymes according to their catalytic mode, producing unsaturated products by β-elimination. An enzyme with an exolytic mode of action performs cleavage at one end of the polymeric chain and usually releases one small oligosaccharide: a mono- or less often a di-saccharide. An enzyme with an endolytic mode of action is able to bind and cleave its substrate anywhere on the polymeric chain, releasing degradation products of various lengths [14]. According to the currently characterized alginate lyases, most of them are endotypes.

In clinical practice, *P. aeruginosa*, a Gram-negative bacteria, is a common opportunistic pathogen that may cause infections of the respiratory system, urinary system, burn wounds, surgeries, skin, and mucous membranes [15]. When infection is inadequately treated and unhealed, *P. aeruginosa* is prone to mucoid transformation, creating a biofilm that wraps around the bacterial body, allowing it to not only evade the patient’s immune system but also to increase antibiotic resistance. Biofilms are highly intricate aggregates in which bacteria are encased within a self-generating matrix of extracellular polymers (EPS). It is a critical strategy for them to survive in the face of unexpected changes in living conditions, including temperature fluctuations and trophic availability [16]. Exopolysaccharides, such as pel, psl, and alginate, are present in the biofilms of P. aeruginosa. Pel is composed of N-acetylgalactosamine and N-acetylglucosamine [17]. Psl is composed of galactose and mannose [18]. By eliminating alginate from biological matrices, alginate lyase was believed to boost the efficacy of antibiotics [19]. 

In this study, a new salt-activated characteristic alginate lyase from the marine bacterium *Tamlana* sp. I1 is identified and characterized, and the effect of the SP on the enzymatic properties was also explored and presented. The SP is discovered to not only influence the expression and purification of the alginate lyase protein, but also several biochemical characteristics of the enzyme. The inhibiting and removing effect of the enzyme on the *P. aeruginosa* biofilm after the removal of the signal peptide are reported as well.

## 2. Results and Discussion

### 2.1. Sequence Analysis of Alyi1/alyI1-1

The marine bacterium *Tamlana* sp. I1 was isolated from sea cucumber gut in the sea cucumber culture pond (Shandong University) in Weihai, China. With a high enzyme activity, it had grown rapidly on a single carbon source with only alginate and effectively degraded alginate. (The results described above have not yet been published.) Gene *alyI1/alyI1-1* in the genome of *Tamlana* sp. I1 was classified as an alginate lyase gene.

Alginate lyase (ALYI1-1) was encoded by the 897-bp gene *alyI1-1* (GeneBank accession number: ON186791). This protein was predicted to contain 298 amino acid residues and a signal peptide of 19 residues. The calculated ALYI1-1 protein had a theoretical isoelectric point (pI) of 4.62 and a theoretical molecular weight (Mw) of 33.23 kDa. The ALYI1 protein (where ALYI1 had 279 amino acid residues after removing the signal peptide) shared the most similarities (30.00%) to AlyA1-II, a characterized PL7 alginate lyase from *Sphingomonas* sp. A1 (Genbank: BAD16656) [20]. The calculated pI of ALYI1 protein was 4.55 and Mw was 32.15 kDa. According to the CAZy database, the PL7 family had six subfamilies. As shown in Figure 1, ALYI1 and other alginate lyases could not be clustered together, so ALYI1 can be considered as the first member of a new subfamily. Moreover, according to the protein sequence alignment through DNAMAN (Figure 2), ALYI1 contains three highly conserved modules of the PL7 family: R(T/S) ELR, catalytic motif (QIH), and Y(F/L) K(L/A) G, consistent with the report by Zhu [21] contributing to the active core of alginate lyase and playing a key role in substrate identification and catalysis [22]. Critical residues Gln181, His172, and Tyr272, also found in ALYI1, were also highly conserved in the characterized structure of PL7 alginate lyases and were implicated in the protein–substrate interaction [22]. Therefore, ALYI1 was inferred to be a novel alginate lyase belonging to a new subfamily of PL7 based on the low similarity, multiple sequence alignment results, and phylogenetic analysis.

Secondary structure analysis was performed by convolution fitting of FT-IR to obtain the area ratios of peak attribution. As shown in Table 1, the proportions of β-sheet, 𝛼-Helix, and random were higher for ALYI1 than for ALYI1-1, while ALYI1-1 had more β-turn. Figure 3 showed the peak area after secondary structure fitting of ALYI1/ALYI1-1. The data suggest that signal peptides affected the secondary structure of proteins. 

### 2.2. Expression, Purification, and Characterization of rALYI1/rALYI1-1

The plate assay and UV absorption technique were used to assess the alginate lyase activity of rALYI1. The findings of the plate experiment revealed that clear transparent circles developed following immersion in iodine (Appendix A), indicating that exogenous expression of rALYI1/rALYI1-1 had been successfully achieved and that rALYI1/rALYI1-1 showed high enzyme activity.

The *alyI1/alyI1-1* genes were effectively cloned, expressed with His-tag in *E. coli* BL21(DE3), and purified using Ni-IDA affinity chromatography. SDS-PAGE examination revealed that the isolated protein appeared as a major band with an Mw of 35 kDa, which was close to the expected Mw (Appendix A). Although rALYI1 and rALYI1-1 had similar enzyme activity, the results of the SDS-PAGE analysis were different. As shown in Figure 3b, the induced band of rALYI1 was much more obvious than the one of rALYI1-1. The above results showed that the SP affected the normal induced expression of the enzyme, while there was no evident effect on enzyme activity.

Over a temperature range of 4–70 °C, the ideal temperature of rALYI1/ rALYI1-1 was found. As shown in Figure 4a, rALYI1 still presented more than 80% enzyme activity at 4–70 °C, and its optimum temperature was 40 °C. Similarly, the ideal temperature of rALYI1-1 was also at 40 °C, and in the temperature range of 50–60 °C, more than 40% enzyme activity was still available and it showed minimal activity at 70 °C, different from the results of rALYI1 with more than 85% activity remaining. 

The thermal stability of rALYI1/rALYI1-1 was determined by detecting the remaining enzyme activity after incubation at different temperatures of 4–70 °C. As shown in Figure 4b, it is common for rALYI1/rALYI1-1 that more than 95% of the enzyme activity maintained after 1h incubation at 4–40 °C and dropped sharply at 45 °C, which indicated that both were thermally stable in the temperature range of 4–40 °C. However, rALYI1 still had about 30% enzyme activity at 45 °C and 50 °C, while rALYI-1 almost presented no activity at 50 °C. The results showed that ALYI1 without signal peptide had a wider optimum temperature range and the enzyme after 1h incubation at various temperatures was more stable compared with rALYI1-1, suggesting that the presence of SP may have an effect on temperature stability and optimum. 

The effects of various pH values on rALYI1 and rALYI1-1 enzyme activities were investigated to determine the optimal pH values. As shown in Figure 4c, rALYI1 stood out in the figure as having a wide optimal pH range at pH 5.0–11.0, where the enzyme activity remained constant. It had the highest enzyme activity at pH 7.6. Whereas, rALYI1-1 (Figure 4c) had a lower enzyme activity of only about 40% in an acidic environment and above 80% at pH 7.0–9.0, its optimum pH was 8.6.

The pH stability of rALYI1 and rALYI1-1 was ensured by incubation in various buffers of different pH (5.0–11.0) at 4 °C for 1 h and the residual activity was assayed. The results showed that rALYI1 had the highest activity at pH 7.6 and it had good stability within 5.0–11.0 with more than 80% activity remaining (Figure 4d). rALYI1-1 (Figure 4d) was unstable under acidic conditions from pH 5.0–8.0, and the enzyme activity started to decrease in a gradient in pH 9.0–11.0. Even though the enzyme activity of rALYI1-1 decreased under acidic and alkaline conditions, it still had 40% residual activity, showing the enzyme had good pH adaptability. It has been reported that the enzyme may be more stable in neutral and slightly alkaline environments, which is consistent with the results in our study.

Thus, the above results expressed that the presence of signal peptide affects the pH of the enzyme, leading to worse pH stability as well as a narrower optimum range. According to the research of currently characterized alginate lyase, the optimum pH was basically at 7.0–8.5, except for Aly2 from *Flammeovirga* sp. MY04, which showed the best activity at pH 6.0 [23]. In our study, although the optimal pH of rALYI1 is 7.6, it may maintain 85% activity over a wide range of 5.0–11.0 as described above. Therefore, it could be inferred that rALYI1 had a wide pH range and was more stable in acidic or alkaline environments, which showed its unique property compared with others.

Various metal ions and surfactants at a final concentration of 1.0 mM were used to evaluate their effects on enzyme activity. As shown in Figure 4e, Mg^2+^, Zn^2+^, and Sr^2+^ had opposite effects on rALYI1 and rALYI1-1, with strong inhibition for the former and better activation for the latter. Na^+^ had significant activation for rALYI1, indicating salt activation properties, which were also explored in detail below. EDTA and SDS had inhibitory effects on both, which is consistent with the reported alginate lyase Aly08 from *Vibrio* sp. SY01 [24] and TsAly6A from *Thalassomonas* sp. LD5 [25]. 

Furthermore, unlike most alginate lyases, Ca^2+^ had an inhibitory effect on rALYI1 and rALYI1-1 enzyme activities. This is consistent with the reported Alg2A from *Flavobacterium* sp. S20 [26] and rSAGL from *Flavobacterium* sp. H63 [27]. Ca^2+^ had been shown to enhance the activity of certain alginate lyases, such as AlyS02 from *Flavobacterium* sp. S02 [28], FlAlyA from *Flavabacterium* sp. UMI-01 [29], Algb from Vibrio sp. W13 [30], Cel32 from *Cellulophaga* sp. NJ-1 [31], and TsAly6A from *Thalassomonas* sp. LD5 [25]. 

The effects of varying Na^+^ concentrations were investigated further (Figure 4f). rALYI1 showed the highest activity at 800 mM NaCl, while the enzyme activity of rALYI1 and ALYI1-1 had no significant change in the range of 1500–5000 mM NaCl addition, indicating that both had salt tolerance. Additionally, rALYI1 showed salt-activating properties when Na^+^ increased enzyme activity in the 100–1000 mm concentration range compared with the control group. The PL7 family contained the following NaCl-activated alginate lyases (A1m [32], rA9mT [33], AlgM4 [34], AlyPM [35], and Aly01 [36]); the enzyme activity from *Pseudoalteromonas* sp. AlyPM increased about six times in 0.5–1.2 M NaCl.

### 2.3. Analysis of Substrate Specificity and Final Products

Various subfamilies of PL7 family had different substrate specificity features. Most alginate lyases of sub-families 3 and 5 were specific or bifunctional, whereas all alginate lyases of the defined subfamily 1 were bifunctional. Glucuronan lyase was the sole enzyme from subfamily 4 that had been studied. Alginate lyases of sub-family 6, including AlyVOA, AlyVOB, AlxM, AlyC3, and A9mT, were all shown to be PM-preferred.

The unsaturated uronic acids were formed by the β-elimination mechanism with alginate, polyG, and polyM as substrates, and then the absorbance value at 235 nm was determined by UV to determine the substrate specificity. As shown in Figure 5a, rALYI1/ rALYI1-1 was active against alginate, polyG, and polyM. The high activity of both toward polyM indicated that rALYI1 and rALYI1-1 were bifunctional alginate lyases with a bias toward polyM.

Thin-layer chromatography (TLC) analysis of alginate, polyM, and polyG degradation products after 24 h was used to evaluate the end products of rALYI1. The primary breakdown products of polyM and alginate were di- and tri-oligosaccharides, according to TLC data (Figure 5b). Electrospray ionization mass spectrometry (ESI-MS) was also used to track the identification of alginate’s final depolymerization products, which included di- and tri-oligosaccharides (Figure 5c). The presence of products of various sizes (DP2 to 6, observed by MS) was a good indication of the endolytic character.

Lineweaver–Burk plots were calculated for the kinetic parameter values of rALYI1/rALYI1-1 at varied doses of sodium alginate as substrate. The specific kinetic parameter values of rALYI1/rALYI1-1 are shown in Table 2. The data in the table show that compared with rALYI1-1, the Km value of rALYI1 was lower, and the Kcat and Kcat/Km values were higher, which indicates that rALYI1 appeared to have a greater substrate binding affinity and catalytic efficiency. The protein amounts of rAlyI1 and rALYI1-1 were 3.90 mg/mL and 2.15 mg/mL. The enzyme activities of both were 2860.27 U/mL and 2720.39 U/mL when sodium alginate was used as the substrate, respectively.

### 2.4. Effect of Alginate Lyase rALYI1 on Biofilm Produced by P. aeruginosa

Alginate lyase has been studied extensively for its therapeutic potential in breaking through the complex matrix of polysaccharides present in *Pseudomonas* biofilms. Several investigations have demonstrated the effectiveness of alginate lyase in breaking biofilm architecture and boosting antibiotic susceptibility. For example, an endolytic enzyme from *Sphingomonas* sp. strain A1 [37], which was highly effective against polyM and has high catalytic activity at pH 8.0 and 70 °C, and an enzyme from the marine bacterium *Pseudoalteromonas* sp. CY24, which had a wider range of substrate specificity and ideal enzyme activity at 40 °C and pH 7.0. The effect on *Pseudomonas* biofilms was investigated using rALYI1, an alginate lyase with high enzyme activity. The enzyme’s capacity to prevent biofilm growth on a polystyrene surface was tested.

The effect of alginate lyase rALYI1 on *P. aeruginosa* biofilm was conducted through the method of a crystal violet staining experiment. In comparison with PBS buffer and denatured rALYI1, the results showed that rALYI1 had an inhibitory effect on *P. aeruginosa* biofilm development. It should be highlighted that rALYI1 has a strong inhibitory effect on biofilm formation, as shown in Figure 6a. Biofilm formation is reduced by 50.53% for *P. aeruginosa*. In addition, the interaction between the rALYI1 enzyme and *P. aeruginosa* biofilms were observed using fluorescent inverted microscopy (Figure 6c,d). Moreover, equal amounts of inactivated enzyme did not show significant inhibition compared with the control, indicating that rALYI1 with higher enzyme activity played a vital role in the inhibition of biofilm formation of *P. aeruginosa*. The composition of the biofilm with bacteria and their exterior matrix might be responsible for the ability to inhibit biofilm formation as previously mentioned. Alginate lyase was discovered to be effective in degrading the primary component of biofilm, alginate, and therefore reducing biofilm development.

The purified enzyme and control groups were incubated with biofilms formed by *P. aeruginosa* for 48 h, respectively, to see if rALYI1 was able to remove the matured biofilm. After staining with crystal violet, the findings (Figure 6b) showed that rALYI1 had greater biofilm destruction, with a degradation rate of roughly 44.98% for *P. aeruginosa*. In addition, a fluorescent inverted microscope was used to study the interaction between the enzyme rALYI1 and the *P. aeruginosa* biofilm (Figure 6e,f).

The influence of ALYI1 on biofilm was also investigated by using the MTT method to detect the content of viable bacteria in the biofilm. The results of the MTT experiment are shown in Figure 7, which supported the above conclusions of the crystal violet experiment.

There have been several reports that alginate lyase could scavenge and inhibit biofilms. For example, AlyP1400 from the marine bacterium *Pseudoalteromonas* sp. 1400, whose catalytic activity has been reported to play an important role in the elimination of biofilms [38]. However, whether the catalytic activity was related to anti-biofilm remains to be investigated. Salt-activating properties were considered to be a relevant factor for the removal of *P. aeruginosa* biofilm by alginate lyase [39], and only with a bifunctional enzyme can the biofilm be removed successfully and efficiently [40].

Nevertheless, the endo- and exo-type cleavage modes of alginate lyase did not differ significantly in the inhibition of *P. aeruginosa* biofilms [37]. The specific factors influencing alginate lyase against *P. aeruginosa* biofilm should be studied in depth. It was essential to be able to effectively assess its efficacy in vivo in the future.

## 3. Materials and Methods

### 3.1. Strains, Plasmids, and Reagents

*Tamlana* sp. I1 was isolated from sea cucumber gut in the sea cucumber culture pond (Shandong University, Weihai China) and preserved in our laboratory, from which the alyI1 and alyI1 -1 genes were cloned. Gene cloning was performed using Escherichia coli DH5α (Sangon Biotech, Shanghai, China). Protein expression was carried out using E. coli BL21 (DE3) (Solarbio, Beijing, China) and the vector pRSFDuet-1 (Invitrogen, America). 3-(4,5)-dimethylthiahiazo (-z-y1)-3,5-di- phenytetrazoliumromide (MTT, Macklin, Shanghai, China). *P. aeruginosa*, the type strain, was preserved in our lab. Other reagents, except where otherwise stated, were of analytical grade or better quality and commercially available.

### 3.2. Cloning, Expression, and Purification of rALYI1/rALYI1-1

Primer 6.0 software (San Francisco, CA, USA) was used to construct the upstream and downstream primer sequences based on the *alyI1* and *alyI1-1* gene sequence and the restriction site of pRSFDuet-1. The primer sequences were as follows: *alyI1*-F (5’-TACTCAGGATCCGACAACCAATGAAG-3’) and *alyI1*-R (5’-TACTCAAAGCTTTTAAAAATGTTCCGTCT-3’), *alyI1-1*-F (5’-TACTCAGGATCCGATGTCCTGTAGT-3’) and *alyI1-1*-R (5’-TACTCAAAGCTTACACATTGAATGCAC-3’) all of which included BamH I and Hind III restriction sites (underlined). After digestion of the PCR fragment with the above restriction enzymes, it was purified and cloned into the pRSFDuet-1expression vector before being converted into *E. coli* BL21 (DE3). The key experimental procedures of expression and purification were described by Yin [41]. The purified fractions were collected and examined with a 10% SDS-PAGE apparatus (Bio-Rad, Hercules, CA, USA). Takara (Beijing, China) provided the protein markers. Measurement of protein concentration was carried out using the Bradford method [42]. Gram’s iodine plate test was used for the rapid and sensitive determination of alginate lyase activity [43].

### 3.3. Sequence Analysis of ALYI1

Functional annotation of predicted proteins was obtained via the National Center for Biotechnology Information (NCBI) (http://www.ncbi.nlm.nih.gov, accessed on 10 April 2021). ExPASy (https://web.expasy.org/compute_pi/, accessed on 10 April 2021) was used to calculate the theoretical isoelectric point (pI) and molecular weight (Mw). The SignalP 5.0 server (http://www.cbs.dtu.dk/services/SignalP/, accessed on 23 December 2021) was used to estimate the signal peptide cleavage site of rALYI1. DNAMAN version 9.0 was used to perform multiple sequence alignments (Lynnon biosoft, San Ramon, CA, USA). MEGA version X was used to perform phylogenetic analysis [44]. The secondary structures of the proteins were detected by Fourier transform infrared spectroscopy (FT-IR, IRSpirit-T, SHIMADZU) [45]. The amide I band in the spectrum (1600–1700 cm^−1^ band) was processed by PeakFit 4.12 (Systat Software Inc., CA, USA) according to the baseline correction, convolution, and fitting. The relevant data obtained for each sub-peak were imported into origin 9.0 (OriginLab, Northampton, MA, USA), and then the data were analyzed.

### 3.4. Enzyme Activity Assay

3,5-dinitrosalicylic acid (DNS) colorimetric assay for detection of alginate lyase activity [46]. The enzyme quantity needed to release 1.0 mol of reductive sugar per minute is the enzyme activity unit (U). Amounts of 50 µL of pH buffer (pH 8.6), 50 µL of alginate substrate, and 50 µL of crude enzyme solution were added to 1.5 ml Eppendorf tubes, respectively, and the process was conducted at 37 °C for 30 min. The experiments were carried out in parallel three times. Immediately after the reaction, 150 µL of DNS was added. It was then boiled for 5 min to stop the reactions. At 540 nm, the absorbance was finally measured.

### 3.5. Characterization of rALYI1/rALYI1-1

Unless otherwise stated, enzyme activity assays were performed using sodium alginate as the test reagent substrate.

#### 3.5.1. Effect of Temperature on Enzyme Activity

The influence of temperature on ALYI1 was studied in the range of 4–70 °C. By incubating the enzyme at 4–70 °C for 1 h, the thermal stability of the enzyme was assessed. Unless otherwise stated, sodium alginate was used as a substrate for the enzyme activity assay. Untreated enzyme activity was defined as 100%.

#### 3.5.2. Effect of pH on Enzyme Activity

Alginate lyase activity was measured across a pH range of 5.0 to 11.0 to determine the optimum pH. The buffer systems employed were 0.2 M Na_2_HPO_4_-0.1M citric acid (pH 5.0–6.6), 0.2 M Na_2_HPO_4_–0.2 M NaH_2_PO_4_ (pH 6.6–7.6), 0.2M Tris 0.1 M HCl buffer (pH 7.6–8.6), and 0.2 M Glycine 0.2 M NaOH buffer (pH 8.6–11.0). Then, 0.5% (*w/v*) sodium alginate as a substrate and an appropriate buffer were mixed and reacted at 40 °C for 30 min. In addition, enzymes were incubated in buffers with different pH values (5.0–11.0) at 4 °C for 12 h to investigate the pH stability.

#### 3.5.3. Effects of Metal ions, Surfactants, and NaCl on Enzyme Activity

The effect of metal ions and surfactants on rALYI1/rALYI1-1 activity was determined according to the residual activity after incubation with various metal ions and surfactants with the final concentration of 1.0 mM. The effects of different concentrations (0–5000 mM) of NaCI were calculated for the enzyme activity with the addition of the same volume of ultrapure water as a control.

### 3.6. Substrate Specificity, Degradation Products, and Kinetic Parameters of rALYI1/rALYI1-1

The preferred substrates for rALYI1/rALYI1-1 were performed based on the UV absorbance at 235 nm with sodium alginate, polyG, and polyM added to a 1.0% (*w/v*) buffer solution (20 mM Na_2_HPO_4_-NaH_2_PO_4_, pH 8.6). The degradation products were detected by the TLC method. After the reaction, the degradation products were conducted on the silica gel plate, with the mobile phase composed of 2:1:l:1 n-butanol, acetic acid, and water for 4 h. The products were dried and the spots appeared after that the aniline diphenylamine solution was sprayed at 85 °C for 5 min. The degradation products were incubated at 37 °C for 24 h and then underwent dialysis. Finally, the degradation products were analyzed by ESI-MS (6230 ESI-TOF MS, Agilent, Palo Alto, CA, USA). The purpose was to further determine the composition of the final product. The specific experimental procedures were referred to as Rui Yin [41].

Hyperbolic regression analysis was used to compute the Km and Vmax values, as previously discussed [47]. The enzyme’s turnover number (Kcat) was also estimated using the ratio of Vmax to enzyme concentration ([E]). Prism 9.0 (GraphPad Software, Inc., La Jolla, CA, USA) and nonlinear regression fitting of the Michaelis–Menten equation, the onset rates of enzymatic activity at different concentrations of sodium alginate were used to determine the onset rates of enzymatic activity and rALYI1/ rALYI1-1 kinetic parameters with different concentrations of sodium alginate as the substrate.

### 3.7. Biofilm Formation and Removal

Biofilm experiments referred to previous research [48]. In this study, *P. aeruginosa* biofilm was used as a model to explore the effect of enzymes on biofilms. The preculture was performed for 24 h at 37 °C on Tryptose Soya Agar (TSA). Single colonies were selected from the plates and placed in Trypticase Soy Broth (TSB) and cultured overnight at 37 °C. The bacterial solution was diluted using the TSB medium to a turbidity of 0.5 and 100 μL of it was added to the 96-well plate. Then, another 100 μL of PBS buffer or purified enzyme was added to the wells. The bacterial suspension was removed from the well after 36 h of incubation and washed. The biofilms were then fixed with 100% methanol and the methanol was removed and washed after 15 min. After the biofilm was air-dried, 200 μL of 1.0% crystal violet 200 μL was added to each well; the biofilm was left for 30 min and then washed with PBS buffer 3 times. Waiting for the pigment biofilm to dry again, 200 μL of 33% acetic acid was added and allowed to stand at 25 °C for 30 min to completely dissolve. Finally, the absorbance at 600 nm was measured.

Biofilm disruption experiments were performed by adding 100 μL of PBS buffer or purified enzyme and incubating at 37 °C for 24 h after initial biofilm formation. Other experimental procedures were the same as those for the inhibition of biofilm formation experiments described above.

The MTT counting method [49] was basically consistent with the above description, except that 20 μL MTT solution (5 mg/mL) was added after washing with PBS and cultured at 37 °C for 6 h. The wells were then carefully siphoned off and 200 μL of dimethyl sulfoxide (DMSO) was added to each well. Finally, the absorbance at 570 nm was determined.

### 3.8. Statistical Analysis

All experiments were conducted independently with at least three replicates on different days, and results were expressed as mean ± standard deviation. To examine the significance of more than two groups, a one-way analysis of variance (ANOVA) was used. *p* < 0.05 was used to evaluate statistical significance. The statistical studies were carried out using the GraphPad Prism version 9.0 statistics program version.

## 4. Conclusions

The main role of the signal peptide is to direct proteins into cellular subcellular organelles containing different membrane structures, influencing protein localization in the cell as well as protein translation modification. However, there were relatively few reports on its biochemical effects on alginate lyase, so we explored the effect of the presence or absence of the SP on the physicochemical properties of the enzyme. In our study, we cloned and characterized alginate lyase with and without SP, respectively. The presence of SP affected the optimum temperature of the enzyme. rALYI1 had a wide range of optimum temperatures and showed good catalytic activity between 4 and 70 °C. In terms of pH, the presence of SP affected the pH range and pH stability, and its optimal range became narrower and more unstable in an acidic environment. Substrate preference analysis showed that both were bifunctional alginate lyases. The above results show that the presence of signal peptide mainly affected enzyme temperature, pH range, and its stability, with no significance on the substrate recognition of the enzyme. This conclusion may provide some additional data to support the role of the SP. Meanwhile, our experiments examined the enzyme action on *Pseudomonas aeruginosa* biofilm.

*P. aeruginosa* biofilms have been reported to be recalcitrant to extreme environments, especially under conditions of extreme temperature, extreme pH, and high salinity [50]. In the present experiments, it was found that rAlyI1 had a significant salt activation property as well as salt tolerance, which might facilitate the breakdown of *P. aeruginosa* biofilm. Several analyses demonstrated that rALYI1 had a clear effect on removing the *P. aeruginosa* biofilm due to its high-efficiency alginate breakdown ability. This result suggested that alginate lyase may be a potential enzyme for the treatment of chronic lung infections caused by *P. aeruginosa*. Nevertheless, an issue that should be still considered was whether the enzyme presents a similar effect in vivo and further research should be undertaken to investigate the specific mechanisms and the functional manner of rALYI1 on biofilms.

## Figures and Tables

**Figure 1 marinedrugs-20-00753-f001:**
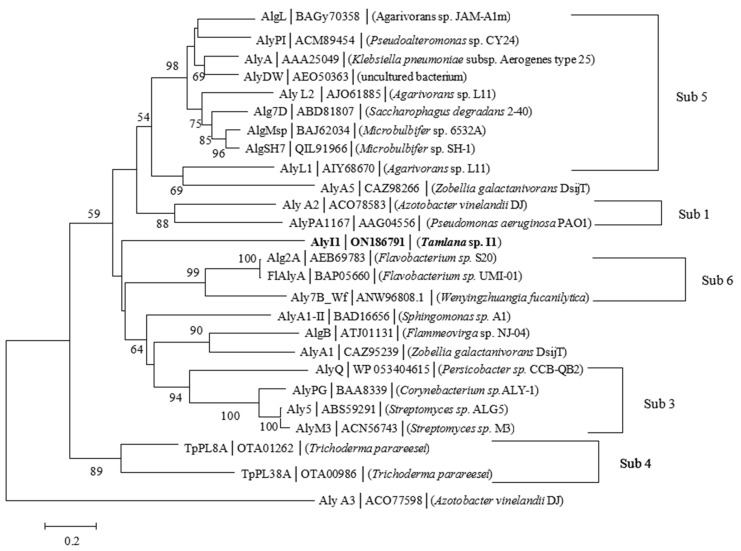
Phylogenetic analysis of AlyIl with other reported alginate lyases. The reliability of the phylogenetic reconstructions was determined by boot-strapping values (1000 replicates). Branch-related numbers are bootstrap values (confidence limits) representing the substitution frequency of each amino acid residue. The species names are indicated along with the accession number of the corresponding alginate lyase sequence. Bootstrap values of 1000 trials are presented in the branching points. Bar, 0.20 substitutions per nucleotide position.

**Figure 2 marinedrugs-20-00753-f002:**
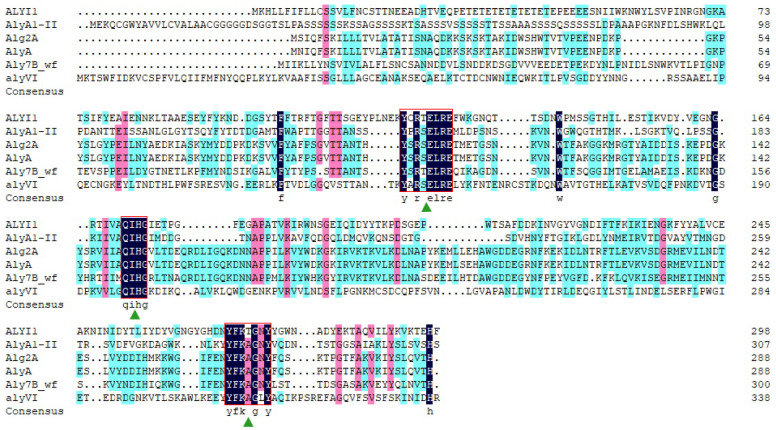
Multiple sequence alignments of alginate lyase AlyI1 and related alginate lyases of the PL7 family. AlyI1 (ON186791) from *Tamlana* sp. I1 in this study, AlyA1-II (BAD16656) from *Sphingomonas* sp. A1, Alg2A(AEB69783) from *Flavobacterium* sp. S20, AlyA (AAA25049) from *Klebsiella pneumonia* subsp Aerogenes type 25, Aly7B_Wf (ANW96808.1) from *Wenyingzhuangia fucanilytica*, and alyVI (AAP45155) from *Vibrio* sp. QY101. The conserved amino acid regions are highlighted with red boxes. The potential residues involved in the catalytic activity in the PL7 family are indicated with green triangles. The depth of the color determines the degree of homology. Dark blue represented conserved amino acids. Pink and cyan represented amino acids with similar functions and structures, with pink representing a higher homology.

**Figure 3 marinedrugs-20-00753-f003:**
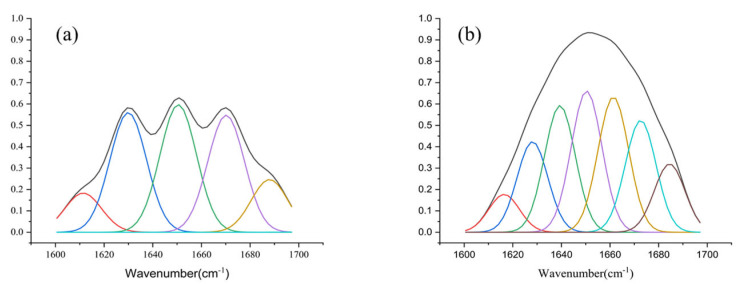
Peak bitmap of secondary structure. (**a**) The fitting results of amide I band of ALYI1. (**b**) The fitting results of the amide I band of ALYI1-1.

**Figure 4 marinedrugs-20-00753-f004:**
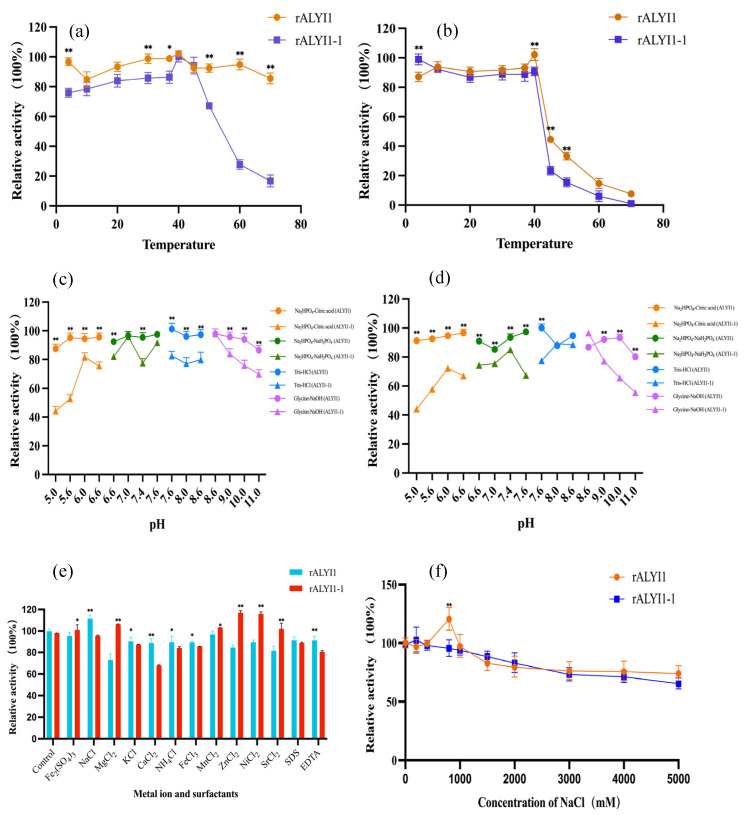
Biochemical characteristics of rALYI1/rALYI1-1. (**a**) Relative activity of rALYI1/rALYI1-1 at different temperatures (4–70 °C). (**b**) Thermostability of rALYI1/rALYI1-1. (**c**) The optimal pH of rALYI1/rALYI1-1. (**d**) pH stability of rALYI1/ rALYI1-1. (**e**) Effects of metal ions and surfactants. (**f**) Effects of NaCl on rALYI1/rALYI1-1. Data are shown as the means ± standard deviation, *n* = 3. * *p* < 0.1; ** *p* < 0.05.

**Figure 5 marinedrugs-20-00753-f005:**
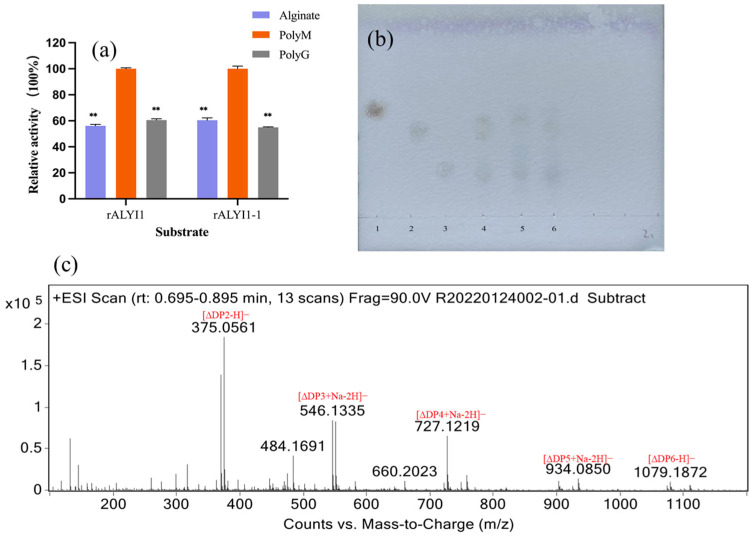
Analysis substrate specificity and final products. (**a**) Substrate specificity of rALYI1/rALYI1-1. (**b**) TLC analysis of rALYI1. Lane 1–3, the purified monomeric sugar, dimer, and trimer standards; lane 4–6, the degradation products of alginate, PM, PG. (**c**) ESI-MS analysis of 24 h enzymatic hydrolysates of rALYI1 using alginate as substrate. The DP2 and DP3 peaks represent a disaccharide and trisaccharide, respectively. DP2 ([∆DP2–H]^–^: *m/z* 375), DP3 ([∆DP3+Na*–*2H]^–^: *m/z* 546), DP4 ([∆DP4+Na–2H]^–^ : *m/z* 727), DP5 ([∆DP3+Na–2H]^–^ : *m/z* 934), and DP6 ([∆DP6–H]^–^: *m/z* 1079). ** *p* < 0.05.

**Figure 6 marinedrugs-20-00753-f006:**
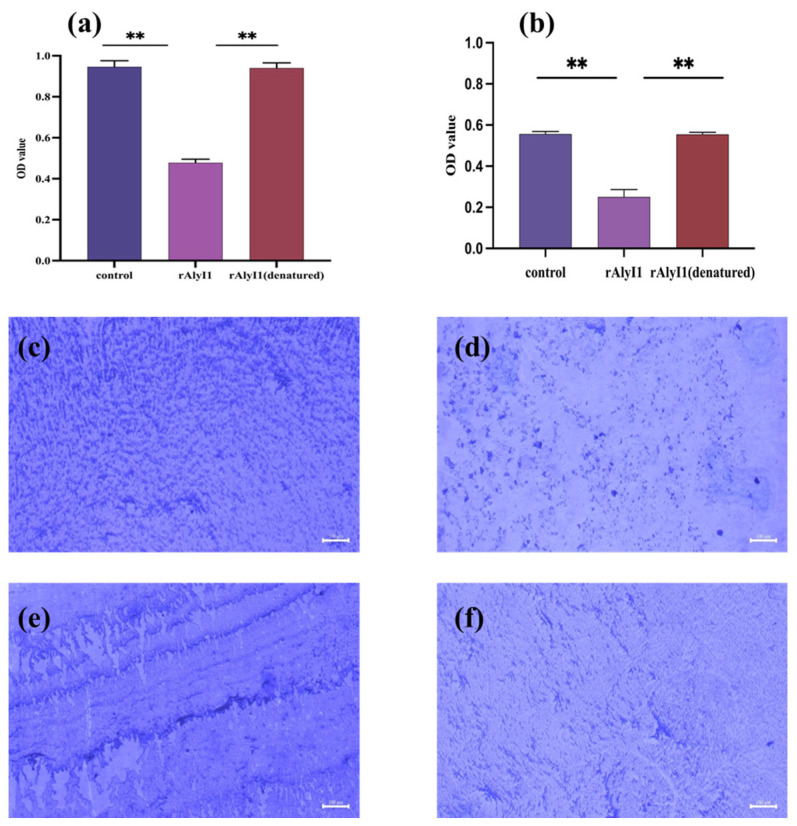
Effect of alginate lyase rALYI1 on biofilm produced by P. aeruginosa. (**a**) Biofilm disruptive activity of rALYI1. (**b**) Prevention of biofilm formation of rALYI1. ** *p <* 0.05. (**c**) Blank control of rAlyI1 inhibiting biofilm formation after crystal violet staining. (**d**) rALYI1 inhibits biofilm formation after crystal violet staining. (**e**) Blank control of rALYI1 destruction of biofilms after crystal violet staining. (**f**) rALYI1 destroys biofilms after crystal violet staining. The scale bar corresponds to 100 μm.

**Figure 7 marinedrugs-20-00753-f007:**
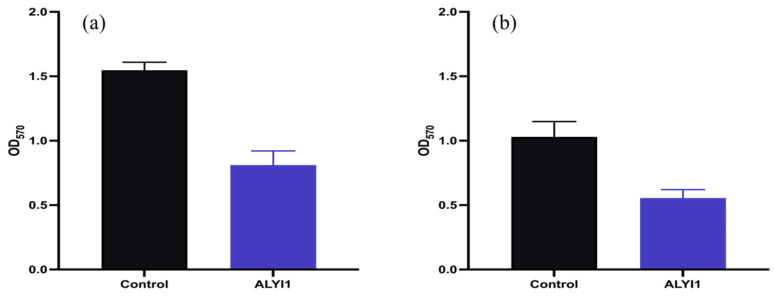
MTT viable bacteria count results. (**a**) Biofilm disruptive activity of rALYI1. (**b**) Prevention of biofilm formation of rALYI1.

**Table 1 marinedrugs-20-00753-t001:** The peak assignment area ratio of ALYI1/ALYI1-1.

Assignment	The Peak Area Ratio of ALYI1 (%)	The Peak Area Ratio of ALYI1-1 (%)
β-sheet	36.8	35.8
β-turn	11.6	25.4
𝛼-Helix	23.6	19.0
Random	28.0	19.8

**Table 2 marinedrugs-20-00753-t002:** The kinetic parameter values of rALYI1/rALYI1-1.

Kinetic Parameter	rALYI1	rALYI1-1
Vmax (μmol min^−1^ mg^−1^)	16.9 ± 0.1	4.1 ± 0.2
Km (mol/L)	0.2 ± 0.1	1.2 ± 0.1
Kcat (s^−1^)	130.1 ± 3.8	5.3 ± 0.3
Kcat/Km(s^−1^ mol^−1^)	541.9 ± 20.2	4.3 ± 0.1

## Data Availability

Not applicable.

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
