# Peer review of "Insights into the Influence of Signal Peptide on the Enzymatic Properties of Alginate Lyase AlyI1 with Removal Effect on Pseudomonas aeruginosa Biofilm"

_marinedrugs, 2022, doi:10.3390/md20120753_

Round 1
Reviewer 1 Report
The manuscript titled “Insights into the Influence of Signal Peptide on the Enzymatic Properties of Alginate Lyase AlyI1 with Removal Effect on Pseudomonas aeruginosa Biofilm” by Zhang et al explores the difference in enzyme properties of AlyI1 with and without signal peptide and shows the effect of the enzyme in inhibiting biofilm formation and eradicating biofilms.
There is a critical lacuna in the main premise of the manuscript. While the authors claim that no one has studied the effect of the signal peptide on enzyme activity although there are studies looking at its effect on folding and secreted expression, they have neglected to consider that misfolding is an obvious factor affecting protein stability and specific enzyme activity. It does not seem to make much sense comparing the two proteins without checking for similar secondary and tertiary structures / protein folding by techniques such as CD spectroscopy or tryptophan fluorescence.
Also, it is not clear if the comparisons have been done with biological replicates and are actually statistically significant. There are no error bars seen in most of the points for the pH stability and optimum pH assay. The last two data points are missing in the thermostability assay for rAlyI1-1. Were they technical or biological replicates (same or different batch of protein prepared from same or different batch of cells? experiment conducted at the same or different time?) which were used for these assays? It is not clear whether there is actually a significant difference between rAlyI1-1 and rAlyI1 in terms of thermostability, optimal pH, pH stability, and effect of metal ions. Also, there is hardly any inhibition observed in the presence of SDS. In my opinion, this section on comparing these enzymes may be removed and just mentioned in passing / put in supplementary data, if deemed necessary.
The rest of the study looking at the effect of the enzyme on biofilms is more interesting and worth reporting. Here, too, the authors could improve their findings by doing cell viability count assays in addition to the crystal violet staining.
Overall, the manuscript is not written very well, and this is especially noticeable in some parts. For instance, lines 175 to 181 need rewriting. Lines 188-189, why use the word “although” in the sentence?
Minor comments
Introduction, line 74: This is incorrect. Alginate lyase is not a biological treatment for cystic fibrosis. It has been studied for exploring its potential use in eradicating P. aeruginosa infections in cystic fibrosis lung.
Line 75-76. Who identified it as the most promising enzyme for the therapy of chronic lung infections caused by Pseudomonas aeruginosa biofilms? Cite appropriate reference, if there is one. DNase, which is currently being used in clinical treatment, is the most useful enzyme so far. PslG also seems to be more efficacious than alginate lysases.
Results and Discussion
Lines 85-88. How did the authors determine that the bacterium had grown rapidly on a single carbon source with only alginate? Does the sea cucumber gut not have any other carbon source? Did the authors actually determine that the bacterium had effectively decomposed alginate within the sea cucumber gut?
Line 94: The sentence is framed incorrectly. It reads as though ALYI1 has only 9 amino acids left after signal peptide removal.
Author Response
We really appreciate your comments. Please see the attachment.

Reviewer 2 Report
The manuscript by Zhang et al. entitled "Insights into the Influence of Signal Peptide on the Enzymatic Properties of Alginate Lyase AlyI1 with Removal Effect on Pseudomonas aeruginosa Biofilm" investigates the influence of the presence of the signal peptide on the biochemical properties of the alginate lyase AlyI1 of Tamlana spI1.
The manuscript is altogether clear and quite well written, with some language errors that need to be corrected (see the minor comments); the references chosen are appropriate and diversified. The figures support well the discourse and are overall of good quality.
The abstract clearly explains the aim of the study and the main results, however the conclusion of it (lines 23-25) seems erroneous since it suggests that the signal peptide is responsible for the inhibition of biofilm formation, while the experiments of biofilm inhibition and degradation have only been performed on the enzyme without its signal peptide.
The introduction and results sections are clear but the latter must be improved according to the major comments listed below. Experimental section is a bit poor and too much written like a protocol; it should be rewritten.
Major comments:
The protein studied here is stated to be referenced in GenBank database under the code ON186791 (line 144) whose query retrieve no result. Please have this fixed.
The figure 2 should be improved by being more consistent in the name of the proteins mentioned. One doesn't know what II1 or AlyA1-II stands for (first and second lines in the alignment). Please explain what do the cyan and dark blue stand for.
Section 2.2 and figure 3a and 3b. The data supporting the production of rAlyI1-1 with a good purity level is missing (Lane for purified rAlyI1-1 on SDS-PAGE). Results of plate assay for rAlyI1-1 are also missing. Please include those data in the manuscript.
The kinetics results in Table 1 are only described (lines 273-278) and not at all discussed. Please discuss these kinetics results.
The biofilm inhibitory capacity of both enzymes must be compared, at least using the crystal violet method.
Section 3.6. This section should be developed. The reference of a previous publication is appropriate but in this case a brief description would be useful to the reader for a good understanding of the experiments carried on. Especially, a brief description of the methods employed to prepare figure 5 is needed (TLC experiment and mass spectrometry).
Minor comments:
Line 12: Please finish this sentence "few studies on its effects on enzymatic properties" by adding "were published" for example.
Line 22: replace " it is also effective" by "rALYI1 is also effective"
Line 37: replace "volume" by "amount"
Line 37: replace "secret" by "secreted"
Line 45: replace "as" by "has"
Lines 54-55: replace "fourteen polysaccharide lyase (PL) families (PL5, 6, 7, 8, 14, 15, 17, 18, 31, 32, 34, 36, 39, and 41) have been classified" by "alginate lyases belong to 14 different polysaccharide lyases families"
Line 58: be more precise, exo mode doesn't always mean release of monosaccharide, it can release a small oligo (disaccharide). Exo mode of action relates to a degradation from one end of the polymeric chain, while an endolytic behaviour relates to a degradation occurring anywhere on the chain.
Lines 68-70: this sentence is too long. Please separate into (at least) 2 sentences.
Line 71: please explicit what are pel and psl.
Line 98: Please be careful with the words used and replace "AlyI-1 and other strains" by "AlyI-1 and other alginate-lyases" since you cannot compare an enzyme and a strain.
Line 99: the word divided is not well applied here, please replace "should be divided into a new subfamily" by "can be considered as the first member of a new subfamily"
Line 100: replace "consisted of" by "contains"
Line 154: replace "higher" by "high"
Line 156: replace "The alyI1/alyI1-1 gene was effectively cloned" by "The alyI1/alyI1-1 genes were effectively cloned"
Figure 4: more consistency in the colors for the 2 proteins might be appreciated for more clarity. For pH graphs, draw one line for each buffer used to highlight a potential effect of the nature of the buffer.
Lines 179-181: I don't understand this sentence
Line 188: please replace "its thermal stability was more stable" by "was more thermally stable"
Line 200: be careful with the use of the adverb extremely: it is in contradiction with what you state after (line 205, good pH adaptability).
Line 227: if you compare behaviour towards NaCl of AlyI1 and AlyI1-1, you have to show the results for AlyI1-1 too.
Line 249: replace "a-elimination" by "β-elimination"
Line 260: the lack of monomer in the products is not sufficient to prove the enzyme has an endo mode of action. However, the presence of products of various sizes (DP2 to 6, observed by MS) is a good indication of the endolytic character.
Line 270: replace "dinosaccharide" by "disaccharide"
Line 325: please add more information about rAlgL enzyme, otherwise one would wonder how it is related to this study.
Lines 325-327: finish this sentence
Line 337: Please add a reference for strain isolation or add a section describing it.
Lines 347-351: BamH I and Hind III restriction sites are stated to be underlined but they are not. Please underline them.
Lines 356, 357, 380, 388, 414, 419: please make real sentences, with verbs. The experimental section should not be written like a protocol.
Line 382: improve English style "Following as described above" is not elegant
Paragraph 3.5.2. (lines 385-390) Please detail which buffer concentration has been used. There is a contradiction about the pH range studied: is it 5-11 (line 385) or 5-12 (line 390)?
Author Response

(The authors gave the same response as above.)

Reviewer 3 Report
I reviewed the manuscript titled "Insights into the influence of signal peptide on the enzymatic properties of alginate lyase AlyI1 with removal effect on Pseudomonas aeruginosa biofilm" and have the following comments: I note that the title can be improved as the words 'removal effect' should be modified. There are also some grammatical errors in some parts of the manuscript though it does not detract from the quality of the work.
The authors examined the role of signal peptides (SP) on the expression and activity profile of an alginate lyase produced by a bacterium from the gut of a sea cucumber. They found that the secretion/production of the Enzyme + SP was lower than the enzyme without the SP. Other key findings were that the pH and temperature range of the enzyme without the SP was much broader. In addition the salt tolerance of this SP-free enzyme was much better. The further studied the effects on biofilms., which they concluded was due to the catalytic activity of the breakdown of alginate in the biofilm.
Overall the methods were presented in a detailed manner which would allow the reader to replicate their experiments. The results were presented clearly but I have to ask if all the sequences shown in Fig 1 are from type strains.
Author Response

(The authors gave the same response as above.)

Reviewer 4 Report
Zhang et al. report on the characterization of the alginate lyase from the marine bacterium Tamlana sp. I1. The gene with and without the encoded signal sequence was cloned and recombinantly expressed and purified from E. coli. Both versions were biochemically and kinetically characterized. Both recombinant enzymes had a similar temperature optimum but the enzyme lacking the signal sequence had a wider optimal temperature range and better thermal stability. In addition, this version was also more stable and acid resistant. Finally, the enzyme lacking the signal sequence displayed a remarkable salt tolerance that suggested it may be a promising enzyme to treat chronic lung infections resulting from Pseudomonas aeruginosa biofilms.
Specific Comments:
(1) Lines 103-5: The sentence needs to be cited.
(2) In the phylogeny on Figure 1, are all of the sequences proven to encode an enzyme with alginate lyase activity?
(3) There are a number of issues with Table 1. First, the Km values are not in the correct units format – they should be in terms of molarity (or millimolarity, etc.). The kcat/Km should also have units. Based on the kcat and Km values provided, I do not know how the kcat/Km value were calculated. Finally, all of the kinetic parameters need error values.
(4) A few grammatical and typographical errors need to be resolved.
Author Response

(The authors gave the same response as above.)

Round 2
Reviewer 2 Report
I thank the authors for their responses. The manuscript has been much improved according to the reviewers' remarks and advice. However a number of minor comments still have to be addressed.
Lines 22: replace "composed of" by "is composed of"
Lines 55-61: the difference between exo- and endo-lytic mode of action is still not clearly explained. An enzyme with an exolytic mode of action performs cleavage at one end of the polymeric chain and usually releases one small oligosaccharide: a mono- or less often a di-saccharide. An enzyme with an endolytic mode of action is able to bind and cleave its substrate anywhere on the polymeric chain, releasing degradation products of various length. Please rephrase this section accordingly, and replace "endocytic" by "endolytic".
Lines 79-80: Replace "but also several the biochemical characteristics" by "but also several biochemical characteristics"
Lines 85-89: please state clearly that these findings are unpublished results
Figure 2: There is still no caption for the pink color, please include a caption for this.
Line 138: Please replace "clear transparent circles developed the following immersion" by "clear transparent circles developed following immersion"
Line 154: Replace "remained" by "remaining"
Line 158 and 162: replace cultivation by incubation
Line 163-164: Replace "suggesting that the presence of SP may have an effect on temperature, especially in the aspect of the optimum temperature" by "suggesting that the presence of SP may have an effect on temperature stability and optimum". As it stands, one could understand that the presence of SP has an impact on temperature itself!
Line 165-170: Please be consistent in writing pH values in all this section. You can choose to write pH 5.0 or pH 5 but stick to this manner of writing afterwards.
Line 227: Replace "method" by "mechanism"
Line 239 : Replace "endocytic" by "endolytic"
Lines 291-2933: The conclusion of the MTT assay is given too quickly. MTT method is used to assess the proportion of live cells, which is related to the biofilm's health. But as it stands, the MTT results is not a direct proof of an inhibition of biofilm formation. It only supports this conclusion.
Lines 359-363: were FTIR data obtained experimentally? If yes, please, explain how, using which equipment, which software, and rephrase the last sentence so that it doesn't look like a protocol.
Line 376: Replace "The influence of temperature on ALYI1 was studied at various temperatures (4–70°C)." by "The influence of temperature on ALYI1 was studied in the range 4–70°C."
Line 396: Replace "substrate solution" by "buffer solution"
Line 399: Replace "unfolded agent of" by "mobile phase composed of"
Line 401: Please explain how the degradation products were desalted. It is of importance because depending on the desalting method, some of the smallest oligosaccharides may be lost.
Line 441: To me, the first sentence of the conclusion is unclear and unnecessary. You may merge it with the following one "The main role of the signal peptide is to direct proteins into…"
Line 447-450: Please rephrase, this sentence is unclear to me.
Line 453: Please rephrase, this sentence is also unclear to me.
Author Response

(The authors gave the same response as above.)

Reviewer 4 Report
Zhang et al. report on the characterization of the alginate lyase from the marine bacterium Tamlana sp. I1. The gene with and without the encoded signal sequence was cloned and recombinantly expressed and purified from E. coli. Both versions were biochemically and kinetically characterized. Both recombinant enzymes had a similar temperature optimum but the enzyme lacking the signal sequence had a wider optimal temperature range and better thermal stability. In addition, this version was also more stable and acid resistant. Finally, the enzyme lacking the signal sequence displayed a remarkable salt tolerance that suggested it may be a promising enzyme to treat chronic lung infections resulting from Pseudomonas aeruginosa biofilms.
Specific Comment:
(1) There are a number of issues with Table 1. First, the Km values still need to be in terms of molarity (or millimolarity, etc.). The units for the kcat/Km in Table 2 should be in sec-1M (or mM)-1. Finally, the kinetic values and errors for all of the kinetic parameters could be the tenth place, not the hundredth.
Author Response

(The authors gave the same response as above.)
